# Breaking the Sample Complexity Barrier to Regret-Optimal Model-Free Reinforcement Learning

**Gen Li**[*]       **Laixi Shi**[†]       **Yuxin Chen**[*]       **Yuantao Gu**[‡]       **Yuejie Chi**[†]
Princeton        CMU          Princeton          Tsinghua              CMU

## Abstract

Achieving sample efficiency in online episodic reinforcement learning (RL) requires optimally balancing exploration and exploitation. When it comes to a finite-horizon episodic Markov decision process with $S$ states, $A$ actions and horizon length $H$, substantial progress has been achieved towards characterizing the minimax-optimal regret, which scales on the order of $\sqrt{H^2 SAT}$ (modulo log factors) with $T$ the total number of samples. While several competing solution paradigms have been proposed to minimize regret, they are either memory-inefficient, or fall short of optimality unless the sample size exceeds an enormous threshold (e.g., $S^6 A^4 \operatorname{poly}(H)$ for existing model-free methods).

To overcome such a large sample size barrier to efficient RL, we design a novel model-free algorithm, with space complexity $O(SAH)$, that achieves near-optimal regret as soon as the sample size exceeds the order of $SA \operatorname{poly}(H)$. In terms of this sample size requirement (also referred to the initial burn-in cost), our method improves — by at least a factor of $S^5 A^3$ — upon any prior memory-efficient algorithm that is asymptotically regret-optimal. Leveraging the recently introduced variance reduction strategy (also called *reference-advantage decomposition*), the proposed algorithm employs an *early-settled* reference update rule, with the aid of two Q-learning sequences with upper and lower confidence bounds. The design principle of our early-settled variance reduction method might be of independent interest to other RL settings that involve intricate exploration-exploitation trade-offs.

## 1 Introduction

Contemporary reinforcement learning (RL) has to deal with unknown environments with unprecedentedly large dimensionality. How to make the best use of samples in the face of high-dimensional state/action space lies at the core of modern RL practice. An ideal RL algorithm would learn to act favorably even when the number of available data samples scales sub-linearly in the ambient dimension of the model, i.e., the number of parameters needed to describe the transition dynamics of the environment. The challenge is further compounded when this task needs to be accomplished with limited memory.

Simultaneously achieving the desired sample and memory efficiency is particularly challenging when it comes to online episodic RL scenarios. In contrast to the simulator setting that permits sampling of any state-action pair, an agent in online episodic RL is only allowed to draw sample trajectories by executing a policy in the unknown Markov decision process (MDP), where the initial states are pre-assigned and might even be chosen by an adversary. Careful deliberation needs to be undertaken when deciding what policies to use to allow for effective interaction with the unknown environment,

---

[*]Department of Electrical and Computer Engineering, Princeton University, Princeton, NJ 08544, USA.

[†]Department of Electrical and Computer Engineering, Carnegie Mellon University, Pittsburgh, PA 15213, USA.

[‡]Department of Electronic Engineering, Tsinghua University, Beijing 100084, China.

35th Conference on Neural Information Processing Systems (NeurIPS 2021).

how to optimally balance exploitation and exploration, and how to process and store the collected information intelligently without causing redundancy.

## 1.1 Regret-optimal model-free RL? A sample size barrier

In order to evaluate and compare the effectiveness of RL algorithms in high dimension, a recent body of works sought to develop a finite-sample theoretical framework to analyze the algorithms of interest, with the aim of delineating the dependency of algorithm performance on all salient problem parameters in a non-asymptotic fashion (Dann et al., 2017; Kakade, 2003). Such finite-sample guarantees are brought to bear towards understanding and tackling the challenges in the sample-starved regime commonly encountered in practice. To facilitate discussion, let us take a moment to summarize the state-of-the-art theory for episodic finite-horizon MDPs with non-stationary transition kernels, focusing on minimizing cumulative regret — a metric that quantifies the performance difference between the learned policy and the true optimal policy — with the fewest number of samples. Here and throughout, we denote by $S$, $A$, and $H$ the size of the state space, the size of the action space, and the horizon length of the MDP, respectively, and let $T$ represent the sample size. In addition, the immediate reward gained at each time step is assumed to lie between 0 and 1.

**Fundamental regret lower bound.** Following the arguments in Jaksch et al. (2010); Auer et al. (2002), the recent works Jin et al. (2018); Domingues et al. (2021) developed a fundamental lower bound on the expected total regret for this setting. Specifically, this lower bound claims that: no matter what algorithm to use, one can find an MDP such that the accumulated regret incurred by the algorithm necessarily exceeds the order of

$$\text{(lower bound)} \qquad \sqrt{H^2 SAT}, \tag{1}$$

as long as $T \geq H^2 SA$.[4] This sublinear regret lower bound in turn imposes a sampling limit if one wants to achieve $\varepsilon$ average regret.

**Model-based RL.** Moving beyond the lower bound, let us examine the effectiveness of model-based RL — an approach that can be decoupled into a model estimation stage (i.e., estimating the transition kernel using available data) and a subsequent stage of planning using the learned model (Jaksch et al., 2010; Azar et al., 2017; Efroni et al., 2019; Agrawal and Jia, 2017; Pacchiano et al., 2020). In order to ensure a sufficient degree of exploration, Zhang et al. (2020b) came up with an algorithm called MVP that blends model-based learning and the optimism principle, which achieves a regret bound[5] $\widetilde{O}\big(\sqrt{H^2 SAT}\big)$ that nearly attains the lower bound (1) as $T$ tends to infinity. Caution needs to be exercised, however, that existing theory does not guarantee the near optimality of this algorithm unless the sample size $T$ surpasses

$$T \geq S^3 AH^4,$$

a threshold that is significantly larger than the dimension of the underlying model. This threshold can also be understood as the initial *burn-in cost* of the algorithm, namely, a sampling burden needed for the algorithm to exhibit the desired performance. In addition, model-based algorithms typically require storing the estimated probability transition kernel, resulting in a space complexity that could be as high as $O(S^2 AH)$ (Azar et al., 2017; Zhang et al., 2020b).

**Model-free RL.** Another competing solution paradigm is the model-free approach, which circumvents the model estimation stage and attempts to learn the optimal values directly (Strehl et al., 2006; Jin et al., 2018; Bai et al., 2019; Yang et al., 2021). In comparison to the model-based counterpart, the model-free approach holds the promise of low space complexity, as it eliminates the need of storing a full description of the model. In fact, in a number of previous works (e.g., Strehl et al. (2006); Jin et al. (2018)), an algorithm is declared to be model-free only if its space complexity is $o(S^2 AH)$ regardless of the sample size $T$.

- *Memory-efficient model-free methods.* Jin et al. (2018) proposed the first memory-efficient model-free algorithm — which is an optimistic variant of classical Q-learning — that achieves a regret bound proportional to $\sqrt{T}$ with a space complexity $O(SAH)$. Compared to the lower bound (1), however, the regret bound in Jin et al. (2018) is off by a factor of $\sqrt{H}$ and hence suboptimal for

---

[4]Given that a trivial upper bound on the regret is $T$, one needs to impose a lower bound $T \geq H^2 SA$ in order for (1) to be meaningful.

[5]Here and throughout, we use the standard notation $f(n) = O(g(n))$ to indicate that $f(n)/g(n)$ is bounded above by a constant as $n$ grows. The notation $\widetilde{O}(\cdot)$ resembles $O(\cdot)$ except that it hides any logarithmic scaling. The notation $f(n) = o(g(n))$ means that $\lim_{n \to \infty} f(n)/g(n) = 0$.

| Algorithm | Regret | Range of sample sizes $T$ that attain optimal regret | Space complexity |
|---|---|---|---|
| UCB-VI (Azar et al., 2017) | $\sqrt{H^2 SAT} + H^4 S^2 A$ | $[S^3 A H^6, \infty)$ | $S^2 A H$ |
| MVP (Zhang et al., 2020b) | $\sqrt{H^2 SAT} + H^3 S^2 A$ | $[S^3 A H^4, \infty)$ | $S^2 A H$ |
| UCB-Q-Hoeffding (Jin et al., 2018) | $\sqrt{H^4 SAT}$ | never | $SAH$ |
| UCB-Q-Bernstein (Jin et al., 2018) | $\sqrt{H^3 SAT} + \sqrt{H^9 S^3 A^3}$ | never | $SAH$ |
| UCB2-Q-Bernstein (Bai et al., 2019) | $\sqrt{H^3 SAT} + \sqrt{H^9 S^3 A^3}$ | never | $SAH$ |
| UCB-Q-Advantage (Zhang et al., 2020c) | $\sqrt{H^2 SAT} + H^8 S^2 A^{\frac{3}{2}} T^{\frac{1}{4}}$ | $[S^6 A^4 H^{28}, \infty)$ | $SAH$ |
| UCB-M-Q (Menard et al., 2021) | $\sqrt{H^2 SAT} + H^4 SA$ | $[SAH^6, \infty)$ | $S^2 A H$ |
| Q-EarlySettled-Advantage **(this work)** | $\sqrt{H^2 SAT} + H^6 SA$ | $[SAH^{10}, \infty)$ | $SAH$ |
| Lower bound (Domingues et al., 2021) | $\sqrt{H^2 SAT}$ | n/a | n/a |

Table 1: Comparisons between prior art and our results for non-stationary episodic MDPs when $T \geq H^2 SA$. The table includes the order of the regret bound, the range of sample sizes that achieve the optimal regret $\widetilde{O}(\sqrt{H^2 SAT})$, and the memory complexity, with all logarithmic factors omitted for simplicity of presentation. The red text highlights the suboptimal part of the respective algorithms.

problems with long horizon. This drawback has recently been overcome in Zhang et al. (2020c) by leveraging the idea of variance reduction (or the so-called "reference-advantage decomposition") for large enough $T$. While the resulting regret matches the information-theoretic limit asymptotically, its optimality in the non-asymptotic regime is not guaranteed unless the sample size $T$ exceeds (see Zhang et al. (2020c, Lemma 7))

$$T \geq S^6 A^4 H^{28},$$

a requirement that is even far more stringent than the burn-in cost imposed by Azar et al. (2017).

- *A memory-inefficient "model-free" variant.* The recent work Menard et al. (2021) put forward a novel sample-efficient variant of Q-learning called UCB-M-Q, which relies on a carefully chosen momentum term for bias reduction. This algorithm is guaranteed to yield near-optimal regret $\widetilde{O}(\sqrt{H^2 SAT})$ as soon as the sample size exceeds $T \geq SA\mathrm{poly}(H)$, which is a remarkable improvement vis-à-vis previous regret-optimal methods (Azar et al., 2017; Zhang et al., 2020c). Nevertheless, akin to the model-based approach, it comes at a price in terms of the space and computation complexities, as the space required to store all bias-value function is $O(S^2 AH)$ and the computation required is $O(ST)$, both of which are larger by a factor of $S$ than other model-free algorithms like Zhang et al. (2020c). In view of this memory inefficiency, UCB-M-Q falls short of fulfilling the definition of model-free algorithms in Strehl et al. (2006); Jin et al. (2018). See Menard et al. (2021, Section 3.3) for more detailed discussions.

A more complete summary of prior results can be found in Table 1.

## 1.2 A glimpse of our contributions

In brief, while it is encouraging to see that both model-based and model-free approaches allow for near-minimal regret as $T$ tends to infinity, they are either memory-inefficient, or require the sample size to exceed a threshold substantially larger than the model dimensionality. In fact, no prior algorithms have been shown to be *simultaneously regret-optimal and memory-efficient* unless

$$T \geq S^6 A^4 \mathrm{poly}(H),$$

which constitutes a stringent sample size barrier constraining their utility in the sample-starved and memory-limited regime. The presence of this sample complexity barrier motivates one to pose a natural question:

> *Is it possible to design an algorithm that accommodates a significantly broader sample size range without compromising regret optimality and memory efficiency?*

In this paper, we answer this question affirmatively, by designing a new model-free algorithm, dubbed as Q-EarlySettled-Advantage, that enjoys the following performance guarantee.

**Theorem 1** (informal)**.** *The proposed* Q-EarlySettled-Advantage *algorithm, which has a space complexity $O(SAH)$, achieves near-optimal regret $\widetilde{O}(\sqrt{H^2SAT})$ as soon as the sample size exceeds $T \geq SA \operatorname{poly}(H)$.*

The proof of this theorem can be found in the full version Li et al. (2021c). As can be seen from Table 1, the space complexity of the proposed algorithm is $O(SAH)$, which is far more memory-efficient than both the model-based approach in Azar et al. (2017) and the UCB-M-Q algorithm in Menard et al. (2021) (both of these prior algorithms require $S^2AH$ units of space). In addition, the sample size requirement $T \geq SA \operatorname{poly}(H)$ of our algorithm improves — by a factor of at least $S^5A^3$ — upon that of any prior algorithm that is simultaneously regret-optimal and memory-efficient. In fact, this requirement is nearly sharp in terms of the dependency on both $S$ and $A$, and was previously achieved only by the UCB-M-Q algorithm at a price of a much higher storage burden.

Let us also briefly highlight the key ideas of our algorithm. As an optimistic variant of variance-reduced Q-learning, Q-EarlySettled-Advantage leverages the recently-introduced reference-advantage decompositions for variance reduction (Zhang et al., 2020c). As a distinguishing feature from prior algorithms, we employ an *early-stopped* reference update rule, with the assistance of two Q-learning sequences that incorporate upper and lower confidence bounds, respectively. The design of our early-stopped variance reduction scheme, as well as its analysis framework, might be of independent interest to other settings that involve managing intricate exploration-exploitation trade-offs.

### 1.3 Related works

We now take a moment to discuss a small sample of other related works; a more extensive discussion is deferred to the supplemental material.

When it comes to online episodic RL (so that a simulator is unavailable), regret analysis is the prevailing analysis paradigm employed to capture the trade-off between exploration and exploitation. A common theme is to augment the original model-free update rule (e.g., the Q-learning update rule) by an exploration bonus, which typically takes the form of, say, certain upper confidence bounds (UCBs) motivated by the bandit literature (Lai and Robbins, 1985; Auer and Ortner, 2010). In addition to the ones in Table 1 for episodic finite-horizon settings, sample-efficient model-free algorithms have been investigated for infinite-horizon MDPs as well (Dong et al., 2019; Zhang et al., 2020b,d; Jafarnia-Jahromi et al., 2020; Liu and Su, 2020; Yang et al., 2021).

The seminal idea of variance reduction was originally proposed to accelerate finite-sum stochastic optimization, e.g., Johnson and Zhang (2013); Gower et al. (2020); Nguyen et al. (2017). Thereafter, the variance reduction strategy has been imported to RL, which assists in improving the sample efficiency of RL algorithms in multiple contexts, including but not limited to policy evaluation (Du et al., 2017; Wai et al., 2019; Xu et al., 2019; Khamaru et al., 2020), RL with a generative model (Sidford et al., 2018a,b; Wainwright, 2019b), asynchronous Q-learning (Li et al., 2020b), and offline RL (Yin et al., 2021).

## 2 Problem formulation

In this section, we formally describe the problem setting. Here and throughout, we denote by $\Delta(\mathcal{S})$ the probability simplex over a set $\mathcal{S}$, and $[M] := \{1, \cdots, M\}$ for any integer $M > 0$.

**Basics of finite-horizon MDPs.** Let $\mathcal{M} = (\mathcal{S}, \mathcal{A}, H, \{P_h\}_{h=1}^H, \{r_h\}_{h=1}^H)$ represent a finite-horizon MDP, where $\mathcal{S} := \{1, \cdots S\}$ is the state space of size $S$, $\mathcal{A} := \{1, \cdots, A\}$ is the action space of size $A$, $H$ denotes the horizon length, and $P_h : \mathcal{S} \times \mathcal{A} \to \Delta(\mathcal{S})$ (resp. $r_h : \mathcal{S} \times \mathcal{A} \to [0, 1]$) represents the probability transition kernel (resp. reward function) at the $h$-th time step, $1 \leq h \leq H$, respectively.

More specifically, $P_h(\cdot \,|\, s, a) \in \Delta(\mathcal{S})$ stands for the transition probability vector from state $s$ at time step $h$ when action $a$ is taken, while $r_h(s, a)$ indicates the immediate reward received at time step $h$ for a state-action pair $(s, a)$ (which is assumed to be deterministic and fall within the range $[0, 1]$). The MDP is said to be non-stationary when the $P_h$'s are not identical across $1 \le h \le H$. A policy of an agent is represented by $\pi = \{\pi_h\}_{h=1}^{H}$ with $\pi_h : \mathcal{S} \to \mathcal{A}$ the action selection rule at time step $h$, so that $\pi_h(s)$ specifies which action to execute in state $s$ at time step $h$. Throughout this paper, we concentrate on deterministic policies.

**Value functions, Q-functions, and Bellman equations.** The value function $V_h^{\pi}(s)$ of a (deterministic) policy $\pi$ at step $h$ is defined as the expected cumulative rewards received between time steps $h$ and $H$ when executing this policy from an initial state $s$ at time step $h$, namely,

$$V_h^{\pi}(s) := \mathop{\mathbb{E}}_{s_{t+1} \sim P_t(\cdot|s_t, \pi_t(s_t)),\, t \ge h} \left[ \sum_{t=h}^{H} r_t\big(s_t, \pi_t(s_t)\big) \,\Big|\, s_h = s \right], \tag{2}$$

where the expectation is taken over the randomness of the MDP trajectory $\{s_t \mid h \le t \le H\}$. The action-value function (a.k.a. the Q-function) $Q_h^{\pi}(s, a)$ of a policy $\pi$ at step $h$ can be defined analogously except that the action at step $h$ is fixed to be $a$, that is,

$$Q_h^{\pi}(s, a) := r_h(s, a) + \mathop{\mathbb{E}}_{\substack{s_{h+1} \sim P_h(\cdot|s,a),\\ s_{t+1} \sim P_t(\cdot|s_t, \pi_t(s_t)),\, t > h}} \left[ \sum_{t=h+1}^{H} r_t\big(s_t, \pi_t(s_t)\big) \,\Big|\, s_h = s, a_h = a \right]. \tag{3}$$

In addition, we define $V_{H+1}^{\pi}(s) = Q_{H+1}^{\pi}(s, a) = 0$ for any policy $\pi$ and any state-action pair $(s, a) \in \mathcal{S} \times \mathcal{A}$. By virtue of basic properties in dynamic programming (Bertsekas, 2017), the value function and the Q-function satisfy the following Bellman equation:

$$Q_h^{\pi}(s, a) = r_h(s, a) + \mathop{\mathbb{E}}_{s' \sim P_h(\cdot|s,a)} \big[ V_{h+1}^{\pi}(s') \big]. \tag{4}$$

A policy $\pi^{\star} = \{\pi_h^{\star}\}_{h=1}^{H}$ is said to be an optimal policy if it maximizes the value function simultaneously for all states among all policies. The resulting optimal value function $V^{\star} = \{V_h^{\star}\}_{h=1}^{H}$ and optimal Q-functions $Q^{\star} = \{Q_h^{\star}\}_{h=1}^{H}$ satisfy

$$V_h^{\star}(s) = V_h^{\pi^{\star}}(s) = \max_{\pi} V_h^{\pi}(s) \qquad \text{and} \qquad Q_h^{\star}(s, a) = Q_h^{\pi^{\star}}(s, a) = \max_{\pi} Q_h^{\pi}(s, a) \tag{5}$$

for any $(s, a, h) \in \mathcal{S} \times \mathcal{A} \times [H]$. It is well known that the optimal policy always exists (Puterman, 2014), and satisfies the Bellman optimality equation:

$$\forall (s, a, h) \in \mathcal{S} \times \mathcal{A} \times [H]: \qquad Q_h^{\star}(s, a) = r_h(s, a) + \mathop{\mathbb{E}}_{s' \sim P_h(\cdot|s,a)} \big[ V_{h+1}^{\star}(s') \big]. \tag{6}$$

**Online episodic RL.** This paper investigates the online episodic RL setting, where the agent is allowed to execute the MDP sequentially in a total number of $K$ episodes each of length $H$. This amounts to collecting

$$T = KH \text{ samples}$$

in total. More specifically, in each episode $k = 1, \dots, K$, the agent is assigned an arbitrary initial state $s_1^k$ (possibly by an adversary), and selects a policy $\pi^k = \{\pi_h^k\}_{h=1}^{H}$ learned based on the information collected up to the $(k-1)$-th episode. The $k$-th episode is then executed following the policy $\pi^k$ and the dynamic of the MDP $\mathcal{M}$, leading to a length-$H$ sample trajectory.

**Goal: regret minimization.** In order to evaluate the quality of the learned policies $\{\pi^k\}_{1 \le k \le K}$, a frequently used performance metric is the cumulative regret defined as follows:

$$\mathsf{Regret}(T) := \sum_{k=1}^{K} \big( V_1^{\star}(s_1^k) - V_1^{\pi^k}(s_1^k) \big). \tag{7}$$

In words, the regret reflects the sub-optimality gaps between the values of the optimal policy and those of the learned policies aggregated over $K$ episodes. A natural objective is thus to design a sample-optimal algorithm, namely, an algorithm whose resulting regret scales optimally in the sample size $T$. Accomplishing this goal requires carefully managing the trade-off between exploration and exploitation, which is particularly challenging in the sample-limited regime.

**Notation.** Before presenting our main results, we take a moment to introduce some convenient notation to be used throughout the remainder of this paper. For any vector $x \in \mathbb{R}^{SA}$ that constitutes certain quantities for all state-action pairs, we shall often use $x(s, a)$ to denote the entry associated with the state-action pair $(s, a)$, as long as it is clear from the context. We shall also let

$$P_{h,s,a} = P_h(\cdot \,|\, s, a) \in \mathbb{R}^{1 \times S} \tag{8}$$

abbreviate the transition probability vector given the $(s, a)$ pair at time step $h$. Additionally, we denote by $e_i$ the $i$-th standard basis vector, with the only non-zero element being in the $i$-th entry and equal to 1.

## 3   Algorithm and theoretical guarantees

In this section, we present the proposed algorithm called Q-EarlySettled-Advantage, as well as the accompanying theory confirming its sample and memory efficiency.

### 3.1   Review: Q-learning with UCB exploration and reference advantage

This subsection briefly reviews the Q-learning algorithm with UCB exploration proposed in Jin et al. (2018), as well as a variant that further exploits the idea of variance reduction (Zhang et al., 2020c). These two model-free algorithms inspire the algorithm design in the current paper.

**Q-learning with UCB exploration** (UCB-Q or UCB-Q-Hoeffding)**.** Recall that the classical $Q$-learning algorithm has been proposed as a stochastic approximation scheme (Robbins and Monro, 1951) to solve the Bellman optimality equation (6), which consists of the following update rule (Watkins, 1989; Watkins and Dayan, 1992):

$$Q_h(s, a) \leftarrow (1 - \eta)Q_h(s, a) + \eta\Big\{ r_h(s, a) + \underbrace{\widehat{P}_{h,s,a}V_{h+1}}_{\text{stochastic estimate of } P_{h,s,a}V_{h+1}} \Big\}. \tag{9}$$

Here, $Q_h$ (resp. $V_h$) indicates the running estimate of $Q_h^\star$ (resp. $V_h^\star$), $\eta$ is the (possibly iteration-varying) learning rate or stepsize, and $\widehat{P}_{h,s,a}V_{h+1}$ is a stochastic estimate of $P_{h,s,a}V_{h+1}$ (cf. (8)). For instance, if one has available a sample $(s, a, s')$ transitioning from state $s$ at step $h$ to $s'$ at step $h + 1$ under action $a$, then a stochastic estimate of $P_{h,s,a}V_{h+1}$ can be taken as $V_{h+1}(s')$, which is unbiased in the sense that

$$\mathbb{E}\big[V_{h+1}(s')\big] = P_{h,s,a}V_{h+1}.$$

To further encourage exploration, the algorithm proposed in Jin et al. (2018) — which shall be abbreviated as UCB-Q or UCB-Q-Hoeffding hereafter — augments the Q-learning update rule (9) in each episode via an additional exploration bonus:

$$Q_h^{\mathsf{UCB}}(s, a) \leftarrow (1 - \eta)Q_h^{\mathsf{UCB}}(s, a) + \eta\big\{ r_h(s, a) + \widehat{P}_{h,s,a}V_{h+1} + b_h \big\}. \tag{10}$$

The bonus term $b_h \geq 0$ is chosen to be a certain upper confidence bound for $(\widehat{P}_{h,s,a} - P_{h,s,a})V_{h+1}$, an exploration-efficient scheme that originated from the bandit literature (Lai and Robbins, 1985; Lattimore and Szepesvári, 2020). The algorithm then proceeds to the next episode by executing/sampling the MDP using a greedy policy w.r.t. the updated Q-estimate. These steps are repeated until the algorithm is terminated.

**Q-learning with UCB exploration and reference advantage** (UCB-Q-Advantage)**.** The regret bounds derived for UCB-Q (Jin et al., 2018), however, fall short of being optimal, as they are at least a factor of $\sqrt{H}$ away from the fundamental lower bound. In order to further shave this $\sqrt{H}$ factor, one strategy is to leverage the idea of variance reduction to accelerate convergence (Johnson and Zhang, 2013; Sidford et al., 2018b; Wainwright, 2019b; Li et al., 2020b). An instantiation of this idea for the regret setting is a variant of UCB-Q based on reference-advantage decomposition, which was put forward in Zhang et al. (2020c) and shall be abbreviated as UCB-Q-Advantage throughout this paper.

To describe the key ideas of UCB-Q-Advantage, imagine that we are able to maintain a collection of reference values $V^{\mathsf{R}} = \{V_h^{\mathsf{R}}\}_{h=1}^H$, which form reasonable estimates of $V^\star = \{V_h^\star\}_{h=1}^H$ and become increasingly more accurate as the algorithm progresses. At each time step $h$, the algorithm adopts the following update rule

$$Q_h^{\mathsf{R}}(s, a) \leftarrow (1 - \eta)Q_h^{\mathsf{R}}(s, a) + \eta\Big\{ r_h(s, a) + \underbrace{\widehat{P}_{h,s,a}\big(V_{h+1} - V_{h+1}^{\mathsf{R}}\big)}_{\text{stochastic estimate of } P_{h,s,a}\left(V_{h+1} - V_{h+1}^{\mathsf{R}}\right)} + \big[\widehat{P_h V_{h+1}^{\mathsf{R}}}\big](s, a) + b_h^{\mathsf{R}} \Big\}. \tag{11}$$

Two ingredients of this update rule are worth noting.

- Akin to the UCB-Q case, we can take $\widehat{P}_{h,s,a}(V_{h+1} - V_{h+1}^{\mathsf{R}})$ to be the stochastic estimate $V_{h+1}(s') - V_{h+1}^{\mathsf{R}}(s')$ if we observe a sample transition $(s, a, s')$ at time step $h$. If $V_{h+1}$ is fairly close to the reference $V_{h+1}^{\mathsf{R}}$, then this stochastic term can be less volatile than the stochastic term $\widehat{P}_{h,s,a}V_{h+1}$ in (10).

- Additionally, the term $\widehat{P_h V_{h+1}^{\mathsf{R}}}$ indicates an estimate of the one-step look-ahead value $P_h V_{h+1}^{\mathsf{R}}$, which shall be computed using a batch of samples. The variability of $\widehat{P_h V_{h+1}^{\mathsf{R}}}$ can be well-controlled through the use of batch data, at the price of an increased sample size.

Accordingly, the exploration bonus term $b_h^{\mathsf{R}}$ is taken to be an upper confidence bound for the above-mentioned two terms combined. Given that the uncertainty of (11) largely stems from these two terms (which can both be much smaller than the variability in (10)), the incorporation of the reference term helps accelerate convergence.

### 3.2 The proposed algorithm: Q-EarlySettled-Advantage

As alluded to previously, however, the sample size required for UCB-Q-Advantage to achieve optimal regret needs to exceed a large polynomial $S^6 A^4$ in the size of the state/action space. To overcome this sample complexity barrier, we come up with an improved variant called Q-EarlySettled-Advantage.

**Motivation: early settlement of a reference value.** An important insight obtained from previous algorithm designs is that: in order to achieve low regret, it is desirable to maintain an estimate of $Q$-function that (i) provides an optimistic view (namely, an over-estimate) of the truth $Q^\star$, and (ii) mitigates the bias $Q - Q^\star$ as much as possible. With two additional optimistic Q-estimates in hand — $Q_h^{\mathsf{UCB}}$ based on UCB-Q, and a reference $Q_h^{\mathsf{R}}$ — it is natural to combine them as follows to further reduce the bias without violating the optimism principle:

$$Q_h(s_h, a_h) \leftarrow \min\left\{ Q_h^{\mathsf{R}}(s_h, a_h),\, Q_h^{\mathsf{UCB}}(s_h, a_h),\, Q_h(s_h, a_h) \right\}. \tag{12}$$

This is precisely what is conducted in UCB-Q-Advantage. However, while the estimate $Q_h^{\mathsf{R}}$ obtained with the aid of reference-advantage decomposition provides great promise, fully realizing its potential in the sample-limited regime relies on the ability to quickly *settle on* a desirable "reference" during the initial stage of the algorithm. This leads us to a dilemma that requires careful thinking. On the one hand, the reference value $V^{\mathsf{R}}$ needs to be updated in a timely manner in order to better control the stochastic estimate of $P_{h,s,a}(V_{h+1} - V_{h+1}^{\mathsf{R}})$. On the other hand, updating $V^{\mathsf{R}}$ too frequently incurs an overly large sample size burden, as new samples need to be accumulated whenever $V^{\mathsf{R}}$ is updated.

Built upon the above insights, it is advisable to prevent frequent updating of the reference value $V^{\mathsf{R}}$. In fact, it would be desirable to stop updating the reference value once a point of sufficient quality — denoted by $V^{\mathsf{R,final}}$ — has been obtained. Locking on a reasonable reference value early on means that (a) fewer samples will be wasted on estimating a drifting target $P_h V_{h+1}^{\mathsf{R}}$, and (b) all ensuing samples can then be dedicated to estimating the key quantity of interest $P_h V_{h+1}^{\mathsf{R,final}}$.

***Remark* 1.** In Zhang et al. (2020c), the algorithm UCB-Q-Advantage requires collecting $\widetilde{O}(SAH^6)$ samples *for each state* before settling on the reference value, which inevitably contributes to the large burn-in cost.

**The proposed Q-EarlySettled-Advantage algorithm.** We now propose a new model-free algorithm that allows for early settlement of the reference value. A few key ingredients are as follows.

- *An auxiliary sequence based on LCB.* In addition to the two optimistic Q-estimates $Q_h^{\mathsf{R}}$ and $Q_h^{\mathsf{UCB}}$ described previously, we intend to maintain another *pessimistic* estimate $Q_h^{\mathsf{LCB}} \leq Q_h^\star$ using the subroutine `update-lcb-q`, based on lower confidence bounds (LCBs). We will also maintain the corresponding value function $V_h^{\mathsf{LCB}}$, which lower bounds $V_h^\star$.

- *Termination rules for reference updates.* With $V_h^{\mathsf{LCB}} \leq V_h^\star$ in place, the updates of the references (lines 15-18 of Algorithm 1) are designed to terminate when

$$V_h(s_h) \leq V_h^{\mathsf{LCB}}(s_h) + 1 \leq V_h^\star(s_h) + 1. \tag{13}$$

---

**Algorithm 1:** Q-EarlySettled-Advantage

---

1 **Parameters:** some universal constant $c_{\mathrm{b}} > 0$ and probability of failure $\delta \in (0, 1)$;

2 **Initialize** $Q_h(s,a), Q_h^{\mathsf{UCB}}(s,a), Q_h^{\mathsf{R}}(s,a) \leftarrow H; V_h(s), V_h^{\mathsf{R}}(s) \leftarrow H; Q_h^{\mathsf{LCB}}(s,a) \leftarrow 0;$
$V_h^{\mathsf{LCB}}(s) \leftarrow 0; N_h(s,a) \leftarrow 0;$
$\mu_h^{\mathsf{ref}}(s,a), \sigma_h^{\mathsf{ref}}(s,a), \mu_h^{\mathsf{adv}}(s,a), \sigma_h^{\mathsf{adv}}(s,a), \delta_h^{\mathsf{R}}(s,a), B_h^{\mathsf{R}}(s,a) \leftarrow 0;$ and $u_{\mathrm{ref}}(s) = \mathsf{True}$ for all
$(s,a,h) \in \mathcal{S} \times \mathcal{A} \times [H].$

3 **for** *Episode $k = 1$* **to** $K$ **do**

4      Set initial state $s_1 \leftarrow s_1^k.$

5      **for** *Step $h = 1$* **to** $H$ **do**

6          Take action $a_h = \pi_h^k(s_h) = \arg\max_a Q_h(s_h, a)$, and draw $s_{h+1} \sim P_h(\cdot \,|\, s_h, a_h)$.
            `// sampling`

7          $N_h(s_h, a_h) \leftarrow N_h(s_h, a_h) + 1; n \leftarrow N_h(s_h, a_h).$ `// update the counter`

8          $\eta_n \leftarrow \frac{H+1}{H+n}.$ `// update the learning rate`

9          $Q_h^{\mathsf{UCB}}(s_h, a_h) \leftarrow \texttt{update-ucb-q}().$ `// run UCB-Q; see Algorithm 2`

10         $Q_h^{\mathsf{LCB}}(s_h, a_h) \leftarrow \texttt{update-lcb-q}().$ `// run LCB-Q; see Algorithm 2`

11         $Q_h^{\mathsf{R}}(s_h, a_h) \leftarrow \texttt{update-ucb-q-advantage}().$ `// estimate` $Q_h^{\mathsf{R}}$`; see Algorithm 2`
         `/* update Q-estimates using all estimates in hand, and update value`
            `estimates`                                                             `*/`

12         $Q_h(s_h, a_h) \leftarrow \min\big\{Q_h^{\mathsf{R}}(s_h, a_h), Q_h^{\mathsf{UCB}}(s_h, a_h), Q_h(s_h, a_h)\big\}.$

13         $V_h(s_h) \leftarrow \max_a Q_h(s_h, a).$

14         $V_h^{\mathsf{LCB}}(s_h) \leftarrow \max\big\{\max_a Q_h^{\mathsf{LCB}}(s_h, a), V_h^{\mathsf{LCB}}(s_h)\big\}.$
         `/* update reference values`                                                  `*/`

15         **if** $V_h(s_h) - V_h^{\mathsf{LCB}}(s_h) > 1$ **then**

16            $V_h^{\mathsf{R}}(s_h) \leftarrow V_h(s_h).$

17         **else if** $u_{\mathrm{ref}}(s_h) = \mathsf{True}$ **then**

18            $V_h^{\mathsf{R}}(s_h) \leftarrow V_h(s_h), \qquad u_{\mathrm{ref}}(s_h) = \mathsf{False}.$

---

Note that $V_h^{\mathsf{R}}$ keeps tracking the value of $V_h$ before it stops being updated. In effect, when the additional condition in lines 15 is violated and thus (13) is satisfied, we claim that it is unnecessary to update the reference $V_h^{\mathsf{R}}$ afterwards, since it is of sufficient quality (being close enough to the optimal value $V_h^\star$) and further drifting the reference does not appear beneficial. As we will make it rigorous shortly, this reference update rule is sufficient to ensure that $|V_h - V_h^{\mathsf{R}}| \leq 2$ throughout the execution of the algorithm, which in turn suggests that the standard deviation of $\widehat{P}_{h,s,a}(V_{h+1} - V_{h+1}^{\mathsf{R}})$ might be $O(H)$ times smaller than that of $\widehat{P}_{h,s,a}V_{h+1}$ (i.e., the stochastic term used in (9) of UCB-Q). This is a key observation that helps shave the addition factor $H$ in the regret bound of UCB-Q.

- *Update rules for $Q_h^{\mathsf{UCB}}$ and $Q_h^{\mathsf{R}}$*. The two optimistic Q-estimates $Q_h^{\mathsf{UCB}}$ and $Q_h^{\mathsf{R}}$ are updated using the subroutine `update-ucb-q` (following the standard Q-learning with Hoeffding bonus (Jin et al., 2018)) and `update-ucb-q-advantage`, respectively. Note that $Q_h^{\mathsf{R}}$ continues to be updated even after $V_h^{\mathsf{R}}$ is no longer updated.

**Q-learning with reference-advantage decomposition.** The rest of this subsection is devoted to explaining the subroutine `update-ucb-q-advantage`, which produces a Q-estimate $Q^{\mathsf{R}}$ based on the reference-advantage decomposition similar to Zhang et al. (2020c). To facilitate the implementation, let us introduce the parameters associated with a reference value $V^{\mathsf{R}}$, which include six different components, i.e.,

$$\big[\mu_h^{\mathsf{ref}}(s,a), \sigma_h^{\mathsf{ref}}(s,a), \mu_h^{\mathsf{adv}}(s,a), \sigma_h^{\mathsf{adv}}(s,a), \delta_h^{\mathsf{R}}(s,a), B_h^{\mathsf{R}}(s,a)\big], \tag{14}$$

for all $(s,a,h) \in \mathcal{S} \times \mathcal{A} \times [H]$. Here $\mu_h^{\mathsf{ref}}(s,a)$ and $\sigma_h^{\mathsf{ref}}(s,a)$ estimate the running mean and 2nd moment of the reference $\big[P_h V_{h+1}^{\mathsf{R}}\big](s,a)$; $\mu_h^{\mathsf{adv}}(s,a)$ and $\sigma_h^{\mathsf{adv}}(s,a)$ estimate the running (weighted) mean and 2nd moment of the advantage $\big[P_h(V_{h+1} - V_{h+1}^{\mathsf{R}})\big](s,a)$; $B_h^{\mathsf{R}}(s,a)$ aggregates the empirical standard deviations of the reference and the advantage combined; and last but not least, $\delta_h^{\mathsf{R}}(s,a)$ is the temporal difference between $B_h^{\mathsf{R}}(s,a)$ and its previous value.

---

**Algorithm 2:** Auxiliary functions

---

1 **Function** `update-ucb-q()`:

2   $Q_h^{\mathsf{UCB}}(s_h, a_h) \leftarrow (1 - \eta_n) Q_h^{\mathsf{UCB}}(s_h, a_h) + \eta_n \Big( r_h(s_h, a_h) + V_{h+1}(s_{h+1}) + c_{\mathsf{b}} \sqrt{\frac{H^3 \log \frac{SAT}{\delta}}{n}} \Big)$.

3 **Function** `update-lcb-q()`:

4   $Q_h^{\mathsf{LCB}}(s_h, a_h) \leftarrow (1 - \eta_n) Q_h^{\mathsf{LCB}}(s_h, a_h) + \eta_n \Big( r_h(s_h, a_h) + V_{h+1}^{\mathsf{LCB}}(s_{h+1}) - c_{\mathsf{b}} \sqrt{\frac{H^3 \log \frac{SAT}{\delta}}{n}} \Big)$.

5 **Function** `update-ucb-q-advantage()`:

  /* update the moment statistics of $V_h^{\mathsf{R}}$                                        */

6   $[\mu_h^{\mathsf{ref}}, \sigma_h^{\mathsf{ref}}, \mu_h^{\mathsf{adv}}, \sigma_h^{\mathsf{adv}}](s_h, a_h) \leftarrow$ `update-moments()`;

  /* update the accumulative bonus and bonus difference                             */

7   $[\delta_h^{\mathsf{R}}, B_h^{\mathsf{R}}](s_h, a_h) \leftarrow$ `update-bonus()`;

8   $b_h^{\mathsf{R}} \leftarrow B_h^{\mathsf{R}}(s_h, a_h) + (1 - \eta_n)\frac{\delta_h^{\mathsf{R}}(s_h, a_h)}{\eta_n} + c_{\mathsf{b}}\frac{H^2 \log \frac{SAT}{\delta}}{n^{3/4}}$;

  /* update the Q-estimate based on reference-advantage decomposition        */

9   $Q_h^{\mathsf{R}}(s_h, a_h) \leftarrow$
  $(1 - \eta_n) Q_h^{\mathsf{R}}(s_h, a_h) + \eta_n \big( r_h(s_h, a_h) + V_{h+1}(s_{h+1}) - V_{h+1}^{\mathsf{R}}(s_{h+1}) + \mu_h^{\mathsf{ref}}(s_h, a_h) + b_h^{\mathsf{R}} \big)$;

10 **Function** `update-moments()`:

11   $\mu_h^{\mathsf{ref}}(s_h, a_h) \leftarrow (1 - \frac{1}{n})\mu_h^{\mathsf{ref}}(s_h, a_h) + \frac{1}{n}V_{h+1}^{\mathsf{R}}(s_{h+1})$; // mean of the reference

12   $\sigma_h^{\mathsf{ref}}(s_h, a_h) \leftarrow (1 - \frac{1}{n})\sigma_h^{\mathsf{ref}}(s_h, a_h) + \frac{1}{n}\big(V_{h+1}^{\mathsf{R}}(s_{h+1})\big)^2$; // 2^{nd} moment of the reference

13   $\mu_h^{\mathsf{adv}}(s_h, a_h) \leftarrow (1 - \eta_n)\mu_h^{\mathsf{adv}}(s_h, a_h) + \eta_n\big(V_{h+1}(s_{h+1}) - V_{h+1}^{\mathsf{R}}(s_{h+1})\big)$; // weighted average of the advantage

14   $\sigma_h^{\mathsf{adv}}(s_h, a_h) \leftarrow (1 - \eta_n)\sigma_h^{\mathsf{adv}}(s_h, a_h) + \eta_n\big(V_{h+1}(s_{h+1}) - V_{h+1}^{\mathsf{R}}(s_{h+1})\big)^2$. // weighted 2^{nd} moment of the advantage

15 **Function** `update-bonus()`:

16   $B_h^{\mathsf{next}}(s_h, a_h) \leftarrow$
  $c_{\mathsf{b}} \sqrt{\frac{\log \frac{SAT}{\delta}}{n}} \Big( \sqrt{\sigma_h^{\mathsf{ref}}(s_h, a_h) - \big(\mu_h^{\mathsf{ref}}(s_h, a_h)\big)^2} + \sqrt{H}\sqrt{\sigma_h^{\mathsf{adv}}(s_h, a_h) - \big(\mu_h^{\mathsf{adv}}(s_h, a_h)\big)^2} \Big)$;

17   $\delta_h^{\mathsf{R}}(s_h, a_h) \leftarrow B_h^{\mathsf{next}}(s_h, a_h) - B_h^{\mathsf{R}}(s_h, a_h)$;

18   $B_h^{\mathsf{R}}(s_h, a_h) \leftarrow B_h^{\mathsf{next}}(s_h, a_h)$.

---

As alluded to previously, the Q-function estimation follows the strategy (11) at a high level. Upon observing a sample transition $(s_h, a_h, s_{h+1})$, we compute the following estimates to update $Q^{\mathsf{R}}(s_h, a_h)$.

- The term $\widehat{P}_{h,s,a}\big(V_{h+1} - V_{h+1}^{\mathsf{R}}\big)$ is set to be $V_{h+1}(s_{h+1}) - V_{h+1}^{\mathsf{R}}(s_{h+1})$, which is an unbiased stochastic estimate of $P_{h,s,a}\big(V_{h+1} - V_{h+1}^{\mathsf{R}}\big)$.

- The term $\big[P_h V_{h+1}^{\mathsf{R}}\big](s, a)$ is estimated via $\mu_h^{\mathsf{ref,R}}$ (cf. line 11). Given that this is estimated using all previous samples, we expect the variability of this term to be well-controlled as the sample size increases (especially after $V^{\mathsf{R}}$ is locked).

- The exploration bonus $b_h^{\mathsf{R}}(s, a)$ is updated using $B_h^{\mathsf{R}}(s_h, a_h)$ and $\delta_h^{\mathsf{R}}(s_h, a_h)$ (cf. lines 7-8 of Algorithm 2), which is a confidence bound accounting for both the reference and the advantage. Let us also explain line 8 of Algorithm 2 a bit. If we augment the notation by letting $b_h^{\mathsf{R},n+1}(s, a)$ and $B_h^{\mathsf{R},n+1}(s, a)$ denote respectively $b_h^{\mathsf{R}}(s, a)$ and $B_h^{\mathsf{R}}(s, a)$ after $(s, a)$ is visited for the $n$-th time, then this line is designed to ensure that

$$\eta_n b_h^{\mathsf{R},n+1}(s, a) + (1 - \eta_n) B_h^{\mathsf{R},n}(s, a) \approx B_h^{\mathsf{R},n+1}(s, a).$$

With the above updates implemented properly, $Q_h^{\mathsf{R}}$ provides the advantage-based update of the Q-function at time step $h$, according to the update rule (11).

### 3.3  Main results

Encouragingly, the proposed Q-EarlySettled-Advantage algorithm manages to achieve near-optimal regret even in the sample-limited and memory-limited regime, as formalized by the following theorem; the proof can be found in the full version Li et al. (2021c).

**Theorem 2.** *Consider any $\delta \in (0,1)$, and suppose that $c_{\mathrm{b}} > 0$ is chosen to be a sufficiently large universal constant. Then there exists some absolute constant $C_0 > 0$ such that Algorithm 1 achieves*

$$\mathsf{Regret}(T) \leq C_0 \left( \sqrt{H^2 SAT \log^4 \frac{SAT}{\delta}} + H^6 SA \log^3 \frac{SAT}{\delta} \right) \tag{15}$$

*with probability at least $1 - \delta$.*

Theorem 2 delivers a non-asymptotic characterization of the performance of our algorithm Q-EarlySettled-Advantage. Several appealing features of the algorithm are noteworthy.

- *Regret optimality.* Our regret bound (15) simplifies to

$$\mathsf{Regret}(T) \leq \widetilde{O}\big(\sqrt{H^2 SAT}\big) \tag{16}$$

  as long as the sample size $T$ exceeds

$$T \geq SA \operatorname{poly}(H). \tag{17}$$

  This sublinear regret bound (16) is essentially optimal, as it coincides with the existing lower bound (1) modulo some logarithmic factor.

- *Sample complexity and substantially reduced burn-in cost.* As an interpretation of our theory (16), our algorithm attains $\varepsilon$ average regret (i.e., $\frac{1}{K}\mathsf{Regret}(T) \leq \varepsilon$) with a sample complexity

$$\widetilde{O}\Big(\frac{SAH^4}{\varepsilon^2}\Big).$$

  Crucially, the burn-in cost (17) is significantly lower than that of the state-of-the-art memory-efficient model-free algorithm (Zhang et al., 2020c) (whose optimality is guaranteed only in the range $T \geq S^6 A^4 \operatorname{poly}(H)$).

- *Memory efficiency.* Our algorithm, which is model-free in nature, achieves a low space complexity $O(SAH)$. This is basically un-improvable for the tabular case, since even storing the optimal Q-values alone takes $O(SAH)$ units of space. In comparison, while Menard et al. (2021) also accommodates the sample size range (17), the algorithm proposed therein incurs a space complexity of $O(S^2 AH)$ that is $S$ times higher than ours.

- *Computational complexity.* An additional intriguing feature of our algorithm is its low computational complexity. The runtime of Q-EarlySettled-Advantage is no larger than $O(T)$, which is proportional to the time taken to read the samples. This matches the computational cost of the model-free algorithm UCB-Q proposed in Jin et al. (2018), and is considerably lower than that of the UCB-M-Q algorithm in Menard et al. (2021) (which has a computational cost of at least $O(ST)$).

## 4  Discussion

In this paper, we have proposed a novel model-free RL algorithm, tailored to online episodic settings, that attains near-optimal regret $\widetilde{O}(\sqrt{H^2 SAT})$ and near-minimal memory complexity $O(SAH)$ at once. Remarkably, the near-optimality of the algorithm comes into effect as soon as the sample size rises above $O(SA \operatorname{poly}(H))$, which significantly improves upon the sample size requirements (or burn-in cost) for any prior regret-optimal model-free algorithm (based on the definition of the model-free algorithm in Jin et al. (2018)). We hope that the method and analysis framework developed herein might inspire further studies regarding how to overcome sample size barriers in other important settings, including model-based RL (Azar et al., 2017), RL for discounted infinite-horizon MDPs (Zhang et al., 2020b), and the case with low-complexity function approximation (Jin et al., 2020; Du et al., 2020; Li et al., 2021b), to name just a few. Additionally, our sample size range is not yet optimal in terms of its dependency on the horizon length $H$. How to tighten this dependency is an important topic that is left for future investigation.

## Acknowledgements

L. Shi and Y. Chi are supported in part by the grants ONR N00014-19-1-2404, NSF CCF-2106778, CCF-2007911 and DMS-2134080. Y. Chen is supported in part by the grants AFOSR YIP award FA9550-19-1-0030, ONR N00014-19-1-2120, ARO YIP award W911NF-20-1-0097, ARO W911NF-18-1-0303, NSF CCF-2106739, CCF-1907661, IIS-1900140 and IIS-2100158, and the Princeton SEAS Innovation Award. Part of this work was done while Y. Chen was visiting the Simons Institute for the Theory of Computing. Y. Gu is supported in part by the grant NSFC-61971266.

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
