# A Further related works

We now take a moment to discuss a small sample of other related works. We limit our discussions primarily to RL algorithms in the tabular setting with finite state and action spaces, which are the closest to our work. The readers interested in those model-free variants with function approximation are referred to Du et al. (2019); Fan et al. (2019); Murphy (2005) and the references therein.

**PAC bounds for synchronous and asynchronous Q-learning.** Q-learning is arguably among the most famous model-free algorithms developed in the RL literature (Watkins and Dayan, 1992; Tsitsiklis, 1994; Jaakkola et al., 1994; Szepesvári, 1997), which enjoys a space complexity $O(SAH)$. Non-asymptotic sample analysis and probably approximately correct (PAC) bounds have seen extensive developments in the last several years, including but not limited to the works of Wainwright (2019a); Even-Dar and Mansour (2003); Beck and Srikant (2012); Chen et al. (2020); Li et al. (2021a) for the synchronous setting (the case with access to a generative model or a simulator), and the works of Even-Dar and Mansour (2003); Beck and Srikant (2012); Qu and Wierman (2020); Li et al. (2020b); Chen et al. (2021) for the asynchronous setting (where one observes a single Markovian trajectory induced by a behavior policy). Finite-time guarantees of other variants of Q-learning have also been developed; partial examples include speedy Q-learning (Azar et al., 2011), double Q-learning (Xiong et al., 2020), variance-reduced Q-learning (Wainwright, 2019b; Li et al., 2020b), momentum Q-learning (Weng et al., 2020), and Q-learning for linearly parameterized MDPs (Wang et al., 2021). This line of works did not account for exploration, and hence the success of Q-learning in these settings heavily relies on the access to a simulator or a behavior policy with sufficient coverage over the state-action space.

**Regret analysis for model-free RL with exploration.** When it comes to online episodic RL (so that a simulator is unavailable), regret analysis is the prevailing analysis paradigm employed to capture the trade-off between exploration and exploitation. A common theme is to augment the original model-free update rule (e.g., the Q-learning update rule) by an exploration bonus, which typically takes the form of, say, certain upper confidence bounds (UCBs) motivated by the bandit literature (Lai and Robbins, 1985; Auer and Ortner, 2010). In addition to the ones in Table 1 for episodic finite-horizon settings, sample-efficient model-free algorithms have been investigated for infinite-horizon MDPs as well (Dong et al., 2019; Zhang et al., 2020b,d; Jafarnia-Jahromi et al., 2020; Liu and Su, 2020; Yang et al., 2021).

**Variance reduction in RL.** The seminal idea of variance reduction was originally proposed to accelerate finite-sum stochastic optimization, e.g., Johnson and Zhang (2013); Gower et al. (2020); Nguyen et al. (2017). Thereafter, the variance reduction strategy has been imported to RL, which assists in improving the sample efficiency of RL algorithms in multiple contexts, including but not limited to policy evaluation (Du et al., 2017; Wai et al., 2019; Xu et al., 2019; Khamaru et al., 2020), RL with a generative model (Sidford et al., 2018a,b; Wainwright, 2019b), asynchronous Q-learning (Li et al., 2020b), and offline RL (Yin et al., 2021).

**Model-based approach.** Model-based RL is known to be minimax-optimal in the presence of a simulator (Azar et al., 2013; Agarwal et al., 2020; Li et al., 2020a), beating the state-of-the-art model-free algorithms by achieving optimality for the entire sample size range (Li et al., 2020a). When it comes to online episodic RL, Azar et al. (2017) was the first work that managed to achieve near-optimal regret (at least for large $T$); in fact, this was also the first result (for any algorithm) matching existing lower bounds for large $T$. The sample efficiency of the model-based approach has subsequently been established for other settings, including but not limited to discounted infinite-horizon MDPs (He et al., 2020), MDPs with bounded total reward (Zanette and Brunskill, 2019; Zhang et al., 2020b), and Markov games (Zhang et al., 2020a).

**Regret lower bound.** Inspired by the classical lower bound argument developed for multi-armed bandits (Auer et al., 2002), the work Jaksch et al. (2010) established a regret lower bound for MDPs with finite diameters (so that for an arbitrary pair of states, the expected time to transition between them is assumed to be finite as long as a suitable policy is used), which has been reproduced in the note Osband and Van Roy (2016) with the purpose of facilitating comparison with Bartlett and Tewari (2009). The way to construct hard MDPs in Jaksch et al. (2010) has since been adapted by Jin et al. (2018) to exhibit a lower bound on episodic MDPs (with a sketched proof provided therein). It was recently revisited in Domingues et al. (2021), which presented a detailed and rigorous proof argument with a different construction.