# OpenReview forum: "Breaking the Sample Complexity Barrier to Regret-Optimal Model-Free Reinforcement Learning"
_NeurIPS.cc/2021/Conference — NeurIPS 2021 Spotlight_

### Official Review · Reviewer_AG59 · 2021-07-15

**Rating:** 7
**Confidence:** 4

**Summary:**

This paper addresses the problem of designing a memory-efficient and sample-efficient algorithm for episodic MDPs with smaller initial warm-up phase. The paper proposes a new model-free algorithm "UCB-Q-Double-Advantage" that extends the optimistic Q-learning and the reference-advantage decomposition for variance reduction using two reference-advantage functions, i.e. the updated reference-advantage function and the old reference-advantage function. Use of these two functions with a well-curated update rule leads to near-optimal regret with SAH^2 warm-up actions, O(SAH) memory complexity, \tilde{O}(optimal x H) sample complexity, and O(T) time complexity. The final performance is O(H) worse than optimal in terms of sample and warm-up complexities.

**Limitations And Societal Impact:**

1. The final performance is O(H) worse than optimal in terms of sample and warm-up complexities. It would be helpful for future developments if authors include a comment on why is it so and whether there is any part of the proof that they think can be improved.
2. The algorithm is in general presented intuitively but the proof technique is not clear in the main paper. The reader keeps on wondering why the presented intuition exactly improves the bound. Is it only the number of reference function updates or is it more than that? If the first is true, then how much than single advantage function?
3. Another point is that authors mentioned similar and even better warm-up complexity and regret is achieved by UCB-M-Q as "at a price of a higher storage burden". The statement seems a bit loose, and as if it is fundamentally necessary to have such larger space complexity to achieve UCB-M-Q's improvements. Explaining whether it is authors intuition or only a way of presenting the result or they think it is fundamental by construction will make the argument rigorous.

**Main Review:**

Originality:
The observations following Equation 13 and its role in reducing updates of the advantage function is original and impactful. Also, the idea of using new and old reference-advantage functions as a proxy of checking the update condition derived from Eq 13 is quite interesting and effective.

Quality:
+ The paper proposes a new algorithm that juggles with warmup and sample complexities as well as space and time complexities quite efficiently with a simple (implementation wise) trick of using a new and an updated reference-advantage function. The proposed trick and its effective use opens up future developments.

- The paper has two main issues. Firstly, though it explains the algorithm and the contributions in light of related work pretty well, the proof technique is never summarised intuitively and briefly. This poses hardness in pointing reasons of limitations and improvements in the proof structure. Secondly, the paper never really implements the algorithm and show its applicability. Such an implementation always increase its usability and testability by readers and practitioners. But this a common issue of most of the papers in this category.


Clarity:
+ In general, the paper presentation is good. The contribution of the paper with respect to the existing literature is well posed in the introduction itself. Also, the formulation of the problem and the existing tricks in literature on which the paper is built are well explained and recognised. The intuition of early settlement and Eqn 13 is also well understood.

- The only problem is that the proof technique is not clear in the main paper. The reader keeps on wondering why the presented intuition exactly improves the warmup or sample complexities. A brief summary of the flow of proof as in App. B.3 along with the points where it deviates from the nearest relative, i.e. UCB-Q-Advantage, will be interesting and insightful. it may also help to understand why still the sample and warmup complexities are suboptimal by H.

Significance:
The paper proposes an effective algorithm design technique that improves performance: the warm-up and sample complexities, and effectiveness: the space and time complexities, of existing algorithms (except UCBMQ, which has better performance but S times more space and time complexities). The use of double advantage functions and their effectiveness may invoke future research in designing more efficient and deployable RL algorithms.

**Time Spent Reviewing:**

3

---

> ### Author Response · Authors · 2021-08-10
> **Thank you for the detailed review and constructive comments!**
>
> We thank gratefully the reviewer for the detailed review and constructive comments. We will revise the work according to the reviewer's suggestions. Below, we provide our response to the reviewer's comments on a point-by-point basis.
>
> ## Clarification on our results in the summary
>
> Thanks for the detailed review and accurate summary. To be precise, our algorithm achieves the optimal regret bound $\tilde{O}(\sqrt{H^2SAT})$  (which is optimal in every parameter up to logarithmic factors) as long as the sample size $T$ exceeds $\tilde{O}(SAH^{10})$ (warm-up sample size). The main contribution is that we signficantly reduce the warm-up sample size to $\tilde{O}(SAH^{10})$, which improves upon the prior regret-optimal algorithm UCB-Q-Advantage (Zhang et al., 2020b) which needs $\tilde{O}(S^6A^4H^{28})$ samples. The final result is only suboptimal in terms of the warm-up complexity in $H$. You may find the detailed comparisons in Table 1.
>
>
> ## Lack of showing technical proof in the main paper.
>
> Owing to the limit of space, we chose to provide more intuition and explanation for the proposed algorithms to help readers to understand and follow, leaving the proof pipeline in the Appendix B.3. As the reviewer suggest, we will add a skeleton of the proof with intuitions in the final version.
>
> ## Lack of practical implementation.
>
> Thank you for the valuable suggestion. We will try our best to conduct numerical experiments and add  results in the final version.
>
> ## Clarity
>
> Thanks for the very useful suggestions. We will add a skeleton of the proof with intuitions in the final version, and point out along the way how the proposed trick leads to an improvement over UCB-Q-Advantage, and where the sub-optimality in terms of $H$ in the warm-up sample size occurs.
>
> ## Sub-optimality of the regret bound.
>
> We will point out where the sub-optimality in terms of $H$ in the warm-up sample size occurs in our proof in the final version.
>
> ## Why the presented intuition exactly improves the bound?
>
> Thanks for raising this question. In the proof, the regret incurs an error caused by updating the reference value, given as $\sum_{h=1}^H\sum_{k=1}^K (V_{h}^{\mathsf{R},k}(s_h^k) -V_{h}^{\mathsf{R},K}(s_h^k))$, where $V_h^{\mathsf{R}, k}$ is the reference value at the $k$th episode. When the reference is updated frequently, this term incurs a high regret on the order of $H^8S^2A^{3/2}T^{1/4}$ as in UCB-Q-Advantage (Zhang et al., 2020b), which is highly sub-optimal. To address this, the update rule (13)  ensures that $V^{\mathsf{R,k}}$ stops being updated as soon as when its value is good enough (close to the optimal value $V^\star$), which reduces the regret caused by the corresponding term to be as low as $O(H^6SA)$. We will describe this high-level intuition in the final version.
>
> ## Is it necessary for UCB-M-Q to incur a large storage burden?
> UCB-M-Q maintains a value function estimate for each state-action-horizon pair, and therefore, necessarily incurs a storage complexity of $S^2AH$. This is formally stated in the origin paper of UCB-M-Q (Menard et al., 2021), see page 6, where we quote *"On the other hand, the space complexity of UCBMQ is the same as the size of the model $HS^2A$."*

---

> > ### Comment · Reviewer_AG59 · 2021-08-26
> > **Reply to Authors' Response**
> >
> > Thank you for your detailed response to reviewers' concerns. I shall be glad to see the final version with the promised clarifications, and thus I retain my affirmative decision.

---

### Official Review · Reviewer_fvnk · 2021-07-16

**Rating:** 7
**Confidence:** 2

**Summary:**

This paper proposes a modified model-free reinforcement learning algorithm for time-inhomogeneous tabular MDP, in which it presents the novel technique of *Double Advantage*. It achieves nearly optimal regret and more importantly, the required sample size of being nearly optimal is significantly smaller than previous works if we require the algorithm to also be space-efficient.

**Limitations And Societal Impact:**

The authors adequately addressed the limitations of their work in Section 4 by presenting them as potential future works. The potential negative social impact is not addressed, which is not so needed for a purely theoretical analysis on tabular MDPs.

**Main Review:**

Overall, I consider the technique of *Double Advantage* developed by this paper is novel in the sense that using two reference value functions circumvents the need of using the unknown $V^\star_h$. Meanwhile, the whole paper is also well-written with lots of intuitive explanation and clear proof sketch in the appendix.

However, I feel hesitant about the significance of its result. In particular, an improvement of space complexity in order of $O(S)$ does not look so valuable compared to a sacrifice of sample complexity in order of $O\left(H^4\right)$.

Questions:
- How does the choice of constant "1" affects the final result in Line 13 of Algorithm 1?
- Is there any more direct intuitive explanation on why the updating rule (13) at the bottom of page 6 decreases the necessary sample size of being optimal?
- At line 221, it says that as long as $V^{\mathsf{R}}_h(s_h)<V^\star_h(s_h)+2$ is satisfied, there will not be any further updates for $V^{\mathsf{R}}_h$. Does this imply that $V^{\mathsf{R}}_h$ decreases monotonically (through *update-ucb-q-advantage*) as episode index $k$ increases? That is, will we have $V^{\mathsf{R}}_h(s_h)\ge V^\star_h(s_h)+2$ in future?
- Although the problem formulation and analysis assumes a time-inhomogeneous MDP, the algorithm itself seems to be applicable for time-homogeneous MDP also. Is there any intuition on how it will perform on a time-homogeneous MDP?
- Does the sample complexity dependency on $S$ and $A$ more important than $H$?

Suggestions on writing:
- $W_h^{\mathsf{R}_i}$ appears without initialization in Algorithm 1.
- It seems better to give a precise mathematical definition of $B^{\mathsf{R}}_h(s, a)$.
- It is probably better to roughly describe in the main text about how you use two reference value functions to avoid the need of using $V^\star_h$, the part of *step 3* in Section B.3.

---

Score increased after rebuttal.

**Time Spent Reviewing:**

7

---

> ### Author Response · Authors · 2021-08-10
> **Thank you for very helpful comments and suggestions!**
>
> We will revise the paper according to the reviewer's suggestions in the final version. In what follows, we provide our response to the reviewer's comments on a point-by-point basis.
>
> ## Regarding the significance of the results.
>
> Thanks for raising this point. We would like to remark that there are a number of RL problems in which the size of the state space dominates and far exceeds the horizon (i.e. $|\mathcal{S}| \gg H$). For example, in the game of Go, the size of the state space could be as astronomically large as $3^{361}$ (since each space has three possible configurations and there are 361 positions), while the horizon is usually much lower (e.g., as small as $150$) [1]. Therefore, alleviating the storage requirement by a factor of $|\mathcal{S}|$ is valuable in such applications.
>
> ## The effect of the constant $1$.
>
> Thanks for the question. Actually, the constant used in line 13 of Algorithm 1 can be replaced by any constant $c = O(1)$ without impacting the final result. In fact, the constant $c = O(1)$ serves as a balance term, which ensures the optimal sample complexity $\tilde{O}(\sqrt{H^2SAT})$ while controlling the warm-up sample complexity to be $\tilde{O} (H^{10}SA)$.  The current paper just picks the value $1$ for simplicity of presentation; we will clarify this point in the revision.
>
> ## The effect of the update rule in (13).
>
> Thanks for the question. In the proof, the regret incurs an error caused by updating the reference value, given as $\sum_{h=1}^H\sum_{k=1}^K (V_{h}^{\mathsf{R},k}(s_h^k) -V_{h}^{\mathsf{R},K}(s_h^k))$, where $V_h^{\mathsf{R}, k}$ is the reference value in the $k$th episode. When the reference is updated frequently, this term incurs a high regret on the order of $H^8S^2A^{3/2}T^{1/4}$ as in UCB-Q-Advantage (Zhang et al., 2020b), which is highly sub-optimal. To address this issue, the update rule (13)  ensures that $V^{\mathsf{R,k}}$ stops being updated as soon as when its value is reasonably close to the optimal value $V^\star$, so as to reduce the regret caused by the corresponding term to be as low as $O(H^6SA)$. We will elaborate on this high-level intuition in the final version.
>
>
> ##  Whether $V_h^R$ is monotonically non-increasing.
>
> Thanks for raising this question. The reviewer is correct that via the update rule in Algorithm 1, $V_h^R(s)$ is monotonically non-increasing as the episode index $k$ increases for all $s\in \mathcal{S}$. This is because $V_h^R(s) = V_h(s)$ after each update, and $V_h(s)$ is monotonically non-increasing by the update rule in line 11-12 in Algorithm 1.
>
> ## Performance on time-homogeneous MDP.
>
> Thanks for the excellent question. Since the optimal regret in the homogeneous MDP is smaller than the inhomogeneous counterpart by a factor of $H$, it requires careful evaluation to verify whether our algorithm can still achieve optimal regret or not.  We will discuss this homogeneous case in the final version.
>
> ## Whether $|\mathcal{S}|, |\mathcal{A}|$ is more important than $H$.
>
> As discussed earlier, the size of the state space could be substantially larger than the other parameters like horizon in a number of RL applications. The size of the action space often depends on applications; for example, in the game of go, $|\mathcal{A}|$ could be comparable to the horizon length, while in some control problems, the action space might sometimes be very large.
>
>
> ## $W_h^R$ appears without initialization in Algorithm 1.
>
> Thanks for raising this point. We will add it at the beginning of Algorithm 1.
>
> ## Adding the definition of $B_h^R(s,a)$ in the main text.
>
> Thanks for the very useful suggestion. In the final version, we will bring back the precise definitions of the auxiliary functions "update-moments'' and "update-bonus'' from the appendix to the main body, which will include the mathematical expression of $B_h^R(s,a)$.
>
> ## In the main text, introducing the effects of the main technical tools to avoid the need of using $V^\star$.
>
> Thanks for this very useful suggestion. We will add more detailed explanation and intuition for this novel technical tool in the main body of the final version. %Considering the space limit, we just briefly introduce the intuition how to avoid the need of using $V^\star$ in line 232-237. As the reviewer suggests, we will add more detailed explanation and intuition for this novel technical tool in the main paper in the final version.
>
>
> [1] Silver, David, et al. "Mastering the game of Go with deep neural networks and tree search." nature 529.7587 (2016): 484-489.

---

> > ### Comment · Reviewer_fvnk · 2021-08-20
> > **Reply to Authors' Response**
> >
> > Thank you very much for your detailed response. My concerns have been well-addressed and I have also increased my score.

---

> > > ### Author Response · Authors · 2021-08-20
> > > **Thanks a lot for your comments**
> > >
> > > Thanks a lot for your comments! Feel free to let us know if you have any further questions.

---

### Official Review · Reviewer_hUgT · 2021-07-19

**Rating:** 8
**Confidence:** 2

**Summary:**

This paper proposes a new memory-efficient model-free algorithm for online episodic RL that achieves near-optimal regret with improved sample size requirement. It exploits reference-advantage decomposition for variance reduction and a double estimator for bias reduction to achieve the improvement.

**Limitations And Societal Impact:**

Considering the specific setting this paper focuses on, the derivation looks solid and I don't see obvious limitations of this work.

**Main Review:**

This work significantly improves the sample complexity barrier for memory-efficient RL algorithms. The use of double estimator to mitigate bias is very interesting and novel as far as I know, and can have implication for other studies.

The paper is very well written and easy to follow. The authors explain the key ideas and intuition of related work and the construction clearly. The theoretical derivations seem correct, though I didn't fully check them all in the supplementary materials.

**Time Spent Reviewing:**

6

---

> ### Author Response · Authors · 2021-08-10
> **Thank you for the positive evaluation!**
>
> We thank gratefully the reviewer for the detailed review and the positive appreciation of our work.

---

### Decision · Program_Chairs · 2021-09-27

**Decision:**

Accept (Spotlight)

**Comment:**

Dear authors,

Following the rebuttal, the reviewers reached a consensus to positively evaluate the contribution.